# Stiff-Eye Syndrome—Anti-GAD Ataxia Presenting with Isolated Ophthalmoplegia: A Case Report

**DOI:** 10.3390/brainsci11070932

**Published:** 2021-07-14

**Authors:** Abel Dantas Belém, Thaís de Maria Frota Vasconcelos, Rafael César dos Anjos de Paula, Francisco Bruno Santana da Costa, Pedro Gustavo Barros Rodrigues, Isabelle de Sousa Pereira, Paulo Roberto de Arruda Tavares, Gabriela Studart Galdino, Daniel Aguiar Dias, Carolina de Figueiredo Santos, Manoel Alves Sobreira-Neto, Pedro Braga-Neto, Paulo Ribeiro Nobrega

**Affiliations:** 1Division of Neurology, Department of Clinical Medicine, Universidade Federal do Ceará, Fortaleza 60430-140, Brazil; abelfdantas@hotmail.com (A.D.B.); thaisvasconcelos53@gmail.com (T.d.M.F.V.); rcap86@hotmail.com (R.C.d.A.d.P.); brsantana93@yahoo.com.br (F.B.S.d.C.); gustavo.rodrigues675@hotmail.com (P.G.B.R.); isabellesousapereira@gmail.com (I.d.S.P.); dr.ptavares@hotmail.com (P.R.d.A.T.); studartgabriela@gmail.com (G.S.G.); daniel.aguiar.dias@gmail.com (D.A.D.); manoelsobreira@ufc.br (M.A.S.-N.); paulo_r_med@yahoo.com.br (P.R.N.); 2Hospital Infantil Albert Sabin, Fortaleza 60410-790, Brazil; carolina.figsantos@gmail.com; 3Centro Universitário Christus, Fortaleza 60190-060, Brazil; 4Center of Health Sciences, Universidade Estadual do Ceará, Fortaleza 60740-903, Brazil

**Keywords:** anti-GAD ataxia, cerebellar ataxia, ophthalmoplegia, stiff person syndrome, autoimmune

## Abstract

Anti-GAD ataxia is one of the most common forms of immune-mediated cerebellar ataxias. Many neurological syndromes have been reported in association with anti-GAD. Ophthalmoparesis has been described in stiff person syndrome. We report a case of anti-GAD ataxia presenting initially with isolated ophthalmoplegia and showing complete resolution after immunotherapy. A 26-year-old male patient presented with ophthalmoparesis characterized by tonic upwards deviation of the right eye. In the following month, he developed progressive ataxia with anti-GAD titers of 1972 UI/mL. After treatment with methylprednisolone and immunoglobulin, there was complete resolution of symptoms and anti-GAD titers decreased. This is the first report of isolated ophthalmoparesis due to tonic eye deviation associated with anti-GAD antibodies without stiff-person syndrome. Tonic eye deviation has been reported in SPS, possibly secondary to continuous discharge in gaze holding neurons in the brainstem (similar to what occurs in spinal motor neurons). With growing evidence for ocular abnormalitites in SPS, anti-GAD associated neurological syndromes should be included in the differential diagnosis of isolated ophthalmoplegia.

## 1. Introduction

In our understanding, since the first description of immune-mediated cerebellar ataxias (IMCAs) by Charcot in 1868, this group of neurological disorders has greatly improved. IMCAs have diverse etiologies such as gluten ataxia, post-infectious cerebellitis, paraneoplastic cerebellar degeneration, opsoclonus myoclonus syndrome, anti-GAD ataxia, and primary autoimmune cerebellar ataxia [1]. The majority of these diseases are associated with autoantibodies targeting neuronal antigens and most are still incompletely described, understood, and are likely underdiagnosed. [2].

Anti-GAD antibodies (Ab) have been associated with multiple neurological syndromes, including stiff-person syndrome (SPS), cerebellar ataxia, and limbic encephalitis, all of which are considered to result from reduced GABAergic transmission [3].

Multiple ocular abnormalities have been described in association with anti-GAD antibodies including nystagmus (up/down-beat, periodic alternating nystagmus) [4], abduction deficits in association with myasthenia and thymoma, slowed and impaired saccade initiation [5], opsoclonus [6], and even ocular flutter [7]. There has been one report of tonic eye deviation associated with SPS [8], but there have been no reports of isolated ophtalmoplegia in association with anti-GAD antibodies.

We report a case of anti-GAD ataxia presenting initially with isolated ophthalmoplegia and showing complete resolution after immunotherapy.

## 2. Case Report

A 26-year-old male patient initially presented with double vision. Symptoms worsened with upward gaze, or when walking, and he noticed intermittent involuntary upward deviation of the right eye. He also reported mild gait imbalance, which improved after closing one eye. He had a previous history of ankylosing spondylitis and type I diabetes mellitus. No previous neurologic symptoms or family history of ataxia were reported.

Upon neurological examination, there was vertical misalignment in the primary gaze, with hypertropia of the right eye (which worsened with upward gaze), along with upbeat nystagmus, which was not gravity-dependent and persisted in the sitting and standing positions (Appendix A). Upon isolated examination of each eye, there was only very mild limitation of elevation of the left eye and persistent involuntary elevation of the right eye, which led us to conclude that a tonic deviation of the right eye was the most likely cause of the primary gaze misalignment. Other ocular movements including adduction, abduction, and depression were normal in both eyes. There was no ataxia on initial examination. Visual acuity, fundoscopy, and campimetry were normal in both eyes. Muscle strength, tonus and reflexes were normal.

After one month, he reported worsening with progressive imbalance, and neurological examination revealed persistent tonic upward deviation of the right eye, gaze-evoked and upbeat nystagmus, and global ataxia, with a Scale for the Assessment and Rating of Ataxia (SARA) score of 11. 

A brain MRI revealed no cerebellar atrophy and no other abnormalities (Figure 1). Cerebrospinal Fluid (CSF) analysis was normal (cells 2/mm^3^, glucose 72 mg/dL, protein 30 mg/dL, and no oligoclonal bands). Complete blood counts, serum vitamin B12, thyroid and liver function tests, urea, electrolytes, glucose, and serum protein electrophoresis were normal. Antinuclear and anti-neutrophil cytoplasmic antibodies were negative. Anti-GAD65 antibody titers were 1972 UI/mL (Anti-GAD ELISA (IgG) Test, Euroimunn, normal range < 10 UI/mL). CT scans of the chest, abdomen and pelvis were negative for neoplasms. Antibodies against Hu/antineuronal nuclear antigen type 1 (ANNA-1), Ri (ANNA-2), Yo (PCA-1), Ma2, CV2/collapsin response mediator protein 5 (CRMP5), and amphiphysin were negative.

A diagnosis of anti-GAD associated ataxia and ophthalmoparesis was made. The patient was treated with methylprednisolone (1 g/day) for five days with only partial improvement. After one month, the symptoms persisted and another course of methylprednisolone was prescribed, this time associated with intravenous immunoglobulin 400 mg/kg/day for five days. 

There was substantial improvement of ophthalmoparesis and resolution of ataxia within one week, and the patient was discharged with continuous azathioprine treatment. Six months after the initial immunoglobulin course, he remained asymptomatic and had complete resolution of ophthalmoparesis (Appendix A). Anti-GAD titers were 20.69 (normal range < 10 UI/mL) six months after immunotherapy and decreased to 7.78 after four years of follow-up. The patient is still asymptomatic with very mild horizontal gaze-evoked nystagmus upon examination.

## 3. Discussion

As far as we know, this is the first report of isolated ophthalmoparesis due to tonic eye deviation associated with anti-GAD antibodies without stiff-person syndrome. Ocular abnormalities were followed by cerebellar ataxia in our patient and had a complete response to immunotherapy. 

It has been proposed that anti-GAD antibodies cause neurological symptoms by inhibiting GABA production in the central nervous system or by inhibiting GABA secretion at nerve terminals [3]. It is not entirely clear why the immunological response directed against GAD causes different clinical phenotypes including SPS, cerebellar ataxia, limbic encephalitis, and epilepsy.

Tonic eye deviation has rarely been reported in conjunction with SPS [8]. It has been hypothesized that tonic eye deviation associated with anti-GAD might be secondary to continuous discharge in gaze holding neurons in the brainstem, similar to what occurs in spinal motor neurons in SPS. The vertical gaze holding neural integrator responsible for maintaining eyes in an eccentric position lies in the interstitial nucleus of Cajal (INC) [9]. Injection of GABA antagonists into vertical and horizontal gaze-holding nuclei impairs gaze-holding mechanisms in experimental animal models [10], leading to the hypothesis that a lack of GABA might result in hyper activation of these structures, possibly causing tonic eye deviation.

Our patient also had gravity-independent upbeat nystagmus and later progressed to horizontal gaze-evoked nystagmus. The gravity independence and associated horizontal nystagmus suggest the dysfunction of neural integrators. A decrease in GABA mediated projections from floccular Purkinje cells to vertical neural integrators in the mesencephalon probably results in gravity-independent upbeat nystagmus, and a decrease in horizontal neural integrators in the medulla results in gaze-evoked nystagmus [11].

The mechanisms underlying GAD-associated cerebellar ataxia are probably also related to a decrease in the pre-synaptic release of GABA, particularly in Purkinje cells [12]. Intracerebellar administration of GAD Abs in rodents has been shown to alter the cerebellar potentiation of corticomotor responses [13]. These findings suggest that reduced release of GABA from pre-synaptic neurons leading to a decrease of downstream inhibitory signals (from Purkinje cells in the case of cerebellar ataxia) is a shared mechanism among GAD-associated neurologic syndromes [12,13].

Recently, a paper demonstrated consistent improvement in saccade velocity after rituximab treatment [14], suggesting that ocular motor abnormalities also improve with immunotherapy and could be a good marker for disease progression and treatment response in SPS. Our patient did not show other signs of stiff person syndrome beyond tonic eye deviation but progressed to cerebellar ataxia. Tonic eye deviation and cerebellar ataxia improved simultaneously, suggesting that some ocular motor abnormalities could be used as markers of treatment response in anti-GAD associated phenotypes other than SPS such as cerebellar ataxia.

Induction and maintenance therapies for anti-GAD associated CA include corticosteroids, IVIg, immunosuppressants, plasmapheresis and rituximab, either alone or in combination, and there appear to be no significant differences in the therapeutic benefits among these immunotherapies [1]. We have chosen to maintain our patient in rituximab, but due to low availability of this drug in our public healthcare system and rapid resolution of symptoms we decided to switch the patient to azathioprine. There was no recurrence of symptoms after two years of follow-up. Anti-GAD titers decreased but remained positive (20.69) after six months of follow-up but were negative (7.78) after four years. A previous study has shown that anti-GAD antibody titers remained high in patients with neurologic disorders, but 4/6 patients in that study had a decrease in previous titers and follow-up was only of 12 months, shorter than the time it took for our patient’s titers to become negative [15].

A limitation of this article is that we did not have access to CSF antibody titers, although high serum GAD antibody titers (>2000 UI/mL) are considered relatively specific of anti-GAD associated neurological syndromes [16]. We also could not asses which epitope was targeted by GAD Abs in our patient. SPS and ataxia patients usually demonstrate recognition of the b78 epitope, while Type 1 diabetes patients demonstrate increased recognition of the b96.11 epitope and low recognition of the b78. GAD65Ab b78 has been show to depress inhibitory synaptic transmission in Purkinje cells with a gradual time course and lasting suppressive effect, while b96.11 causes only a transient depression of transmission. This probably explains why patients presenting GAD-Abs with reactivity to the b78 epitope tend to present with cerebellar ataxia [17].

## 4. Conclusions

In conclusion, we suggest that ocular motor abnormalities should be evaluated in all patients with cerebellar ataxia or suspected SPS or anti-GAD associated neurologic syndromes, as they can yield important clues to diagnosis. Additionally, with growing evidence for ocular abnormalities in SPS, anti-GAD associated neurological syndromes should be included in the differential diagnosis of isolated ophthalmoplegia.

## Figures and Tables

**Figure 1 brainsci-11-00932-f001:**
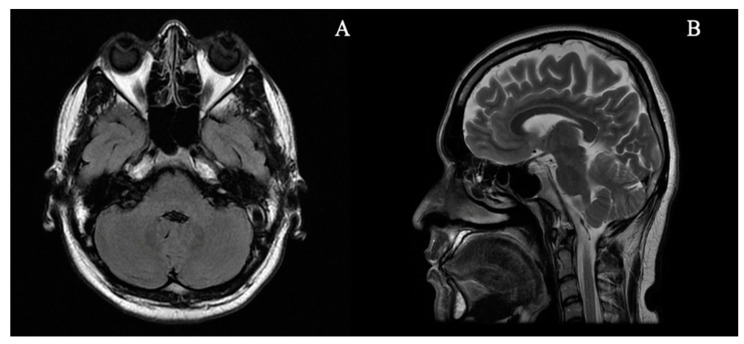
Magnetic Resonance Imaging of the brain showing no significant abnormalities and no cerebellar atrophy. (**A**) Axial Flair weighted image. (**B**) Sagittal T2 weighted image.

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
