# Peer review of "Stiff-Eye Syndrome—Anti-GAD Ataxia Presenting with Isolated Ophthalmoplegia: A Case Report"

_brainsci, 2021, doi:10.3390/brainsci11070932_

Round 1

Reviewer 1 Report

The authors described a patient with anti-GAD ataxia presenting initially with isolated ophthalmoplegia due to tonic eye deviation. He responded to immunotherapy, and this case report would be informative.

The authors should clarify which technique was used to measure GAD antibody.

Is the below 20 UI/ml of anti-GAD titers a negative test result? Please indicate cut-off value of this test.

Did authors check for CSF anti-GAD antibody titers? How about anti-GAD antibody CSF/serum index?

Some previous reports have shown anti-GAD antibody-positive SPS to be associated with neoplasm. Did authors check for tumors, thymoma?

Didn’t the neuro-ophthalmologic examination findings change with the emergence of cerebellar ataxia?

Previous report showed that anti-GAD antibody titer was less likely to decrease even in cases of anti-GAD antibody-positive neurologic disorders with good response to immunotherapy (J Neuroimmunol 2018;317:84-89). Did authors follow anti-GAD antibody titer during the course?

In the Discussion section the authors say that tonic eye deviation associated with anti-GAD might be secondary to continuous discharge in gaze holding neurons in the brainstem. What is the mechanism of his cerebellar ataxia?

Author Response

Response to Reviewer 1 Comments

Point 1: The authors should clarify which technique was used to measure GAD antibody.

Response 1: Very important comment. It was Anti-GAD ELISA (IgG) Test, Euroimunn, to the 65kd isoform. It is now stated in the case report.

Point 2: Is the below 20 UI/ml of anti-GAD titers a negative test result? Please indicate cut-off value of this test.

Response 2: Thank you for pointing that out. We have reviewed actual test values and <10UI/mL is the lower threshold for anti-GAD positivity in our test, as is now indicated in the manuscript. Thank you for this contribution.

Point 3: Did authors check for CSF anti-GAD antibody titers? How about anti-GAD antibody CSF/serum index?

Response 3: Unfortunately we were not able to perform tests for anti-GAD in CSF due to scarce resources. We have now stated that in the limitations section of our paper. However, high anti-GAD Abs (>2000) associated with a typical clinical picture are rarely seem in Type 1 diabetes patients and are relatively specific for neurological syndromes associated with anti-GAD.

 Point 4: Some previous reports have shown anti-GAD antibody-positive SPS to be associated with neoplasm. Did authors check for tumors, thymoma?

Response 4: Very important point. We did search for neoplasms, including thymoma, with a chest/abdomen/pelvis CT, which was negative. It is now stated in the text. Thank you very much.

Point 5: Didn’t the neuro-ophthalmologic examination findings change with the emergence of cerebellar ataxia?

Response 5: This is a very interesting question. Indeed, a gaze-evoked nystagmus appeared, but tonic eye deviation persisted. It is now stated in the manuscript. Thank you for this valuable contribution.

Point 6: Previous report showed that anti-GAD antibody titer was less likely to decrease even in cases of anti-GAD antibody-positive neurologic disorders with good response to immunotherapy (J Neuroimmunol 2018;317:84-89). Did authors follow anti-GAD antibody titer during the course?

Response 6: Thank you for this very important comment. We did follow anti-GAD titers. They were 1972 in 2014 at initial presentation of symptoms, dropped to 20,69 (still positive in our method) six months after immunotherapy, and were 7,78 in 2018 (four years after initial presentation). We have now included this information in the case report and a discussion subject relating to the article cited above, as follows: Anti-GAD titers decreased but remained positive (20,69) after 6 months of follow-up, but were negative (7,78) after 4 years. A previous study has shown that anti-GAD antibody titers remained high in patients with neurologic disorders, but 4/6 patients in that study had a decrease in previous titers and follow-up was only of 12 months, shorter than the time it took for our patients titers to become negative. 

Point 7: In the Discussion section the authors say that tonic eye deviation associated with anti-GAD might be secondary to continuous discharge in gaze holding neurons in the brainstem. What is the mechanism of his cerebellar ataxia??

Response 7: Very interesting point. Thank you very much for bringing that up. It is now stated in the discussion section: “The mechanisms underlying GAD-associated cerebellar ataxia are probably also related to a decrease in pre-synaptic release of GABA, particularly in Purkinje cells. Intracerebellar administration of GAD Abs in rodents has been shown to alter the cerebellar potentiation of corticomotor responses. These findings suggest that reduced release of GABA from pre-synaptic neurons leading to a decrease of downstream inhibitory signals (from Purkinje cells in the case of cerebellar ataxia) is a shared mechanism among GAD-associated neurologic syndromes”

Reviewer 2 Report

The authors report on a case of tonic eye deviation associated with anti-GAD antibodies.

There are several issues which need improvements:

-in the introduction, the first sentence should be associated with a review on IMCAs. There are several recent reviews on the topic (J Mov Disord. 2021 Jan;14(1):10-28 )

-please confirm the list of auto-antibodies which have been searched for

-please explain the gravity-dependence in this patient (Cerebellum. 2019 Apr;18(2):287-290)

-the authors have skipped the discussion on epitopes (Front Behav Neurosci. 2015 Mar 27;9:78.). Did the authors look for the epitope targetted by anti-GAD Ab?

-please explain the choice of immunomodulators

Author Response

Response to Reviewer 2 Comments

Point 1: In the introduction, the first sentence should be associated with a review on IMCAs. There are several recent reviews on the topic (J Mov Disord. 2021 Jan;14(1):10-28)

Response 1: We entirely agree. This is a great suggestion. We have started our introduction with some parts of this article and included it in our reference list. We believe it to be a major improvement to our paper, thank you very much.

Point 2: Please confirm the list of auto-antibodies which have been searched for

Response 2: We have searched for Hu/antineuronal nuclear antigen type 1 (ANNA-1), Ri (ANNA-2), Yo (PCA-1), Ma2, CV2/collapsin response mediator protein 5 (CRMP5), amphiphysin and antiglutamic acid decarboxylase 65 (GAD65). All other Abs were negative. It has now been stated in the case report. Thank you very much for this suggestion.

Point 3: Please explain the gravity-dependence in this patient (Cerebellum. 2019 Apr;18(2):287-290) 

Response 3: Very interesting discussion, thank you. It is now stated in the case report: “…along with upbeat nystagmus, which was not gravity-dependent, persisting in the sitting and standing positions”. And in discussion: “Our patient also had gravity-independent upbeat nystagmus and later progressed to horizontal gaze-evoked nystagmus. The gravity independence and associated horizontal nystagmus suggest dysfunction of neural intergrators. A decrease in GABA mediated projections from floccular Purkinje cells to vertical neural integrators in the mesenceph-alon probably results in gravity-independent upbeat nystagmus, and to horizontal neural integrators in medulla results in gaze-evoked nystagmus.”  

 Point 4: The authors have skipped the discussion on epitopes (Front Behav Neurosci. 2015 Mar 27;9:78.). Did the authors look for the epitope targetted by anti-GAD Ab?

Response 4: Thank you for the comment. Unfortunately we could not look for the epitope targeted, as we lacked the specific tests to do so. We also included the following paragraph in the limitations section in discussion citing information from the reference provided by the reviewer: “We also could not asses which epitope was targeted by GAD Abs in our patient. SPS and ataxia patients usually demonstrate recognition of the b78 epitope, while Type 1 diabetes patients demonstrate increased recognition of the b96.11 epitope and low recognition of the b78. GAD65Ab b78 has been show to depress inhibitory synaptic transmission in Purkinje cells with a gradual time course and lasting suppressive effect, while b96.11 causes only a transient depression of transmission. This probably explains why patients presenting GAD-Abs with reactivity to the b78 epitope tend to present with cerebellar ataxia[17].”

Point 5: please explain the choice of immunomodulators

Response 5: According to Mitoma et al, both the induction and maintenance therapies include corticosteroids, IVIg, immunosuppressants, plasmapheresis, and rituximab, either alone or in combination and there appear to be no significant differences in the therapeutic benefits among the above immunotherapies. We have chosen to maintain our patient in rituximab, but due to low availability of this drug in our public healthcare system and rapid resolution of patients symptoms we decided to switch the patient to azathioprine as a corticoid sparing drug, as the patient has type I diabetes mellitus. It is now stated in discussion. Thank you very much for this remark.

Round 2

Reviewer 1 Report

This version is good. This paper is useful and informative for readers.

Reviewer 2 Report

The authors have replied to the queries